# The typhoid Mary legacy: Genomic epidemiology uncovers contemporary carriage dynamics across two decades of enteric fever surveillance in England and Wales

Alice M. Nisbet[1,2,3], Iman Mohamed[4], Neville Q. Verlander[5], David Powell[6], Emma V. Waters[1,3], Andrew Nelson[7], Hilary Kirkbride[4], Gemma C. Langridge[1,3], Marie A. Chattaway [6,8,9]*

1 Microbes and Food Safety, Quadram Institute Bioscience, Norwich, United Kingdom, 2 School of Medicine, University of East Anglia, Norwich, United Kingdom, 3 Centre for Microbial Interactions, Norwich Research Park, Norwich, United Kingdom, 4 Travel Health and International Health Regulations Team, United Kingdom Health Security Agency, London, United Kingdom, 5 Statistics, Modelling and Economics Department, United Kingdom Health Security Agency, London, United Kingdom, 6 Gastrointestinal Bacteria Reference Unit, United Kingdom Health Security Agency, London, United Kingdom, 7 Gastrointestinal, Emerging and Zoonotic Infections, Public Health Wales, Cardiff, United Kingdom, 8 Health Protection Research Unit in Gastrointestinal Infection, University of East Anglia, Norwich, United Kingdom, 9 Health Protection Research Unit in Public Health Genomics, University of Birmingham, Birmingham, United Kingdom

* marie.chattaway@ukhsa.gov.uk

## Abstract

Modern knowledge regarding 'healthy bacterial carriers' traces back to the legacy of typhoid Mary, who unknowingly infected >50 people with enteric fever. Yet, the intricacies of typhoidal *Salmonella* carriage, inclusive of *Salmonella* Typhi and Paratyphi A/B, remain largely unknown. Using data collected by the United Kingdom Health Security Agency's *Salmonella* reference laboratory and enhanced surveillance, we examined cases of enteric fever in England and Wales between 2004–2023, to track disease trends and identify carriers. In total, 8,297 cases of enteric fever were reported during the study period, including concerning rises post-pandemic that are not linked to emerging bacterial strains. 92% of cases reported travel, mainly to Southern Asia, and disease was most prevalent in more deprived communities. Around 2.7% of cases failed to clear after three weeks, constituting carriage, but only 0.1% persisted over a year. Odds of carriage were significantly increased in patients aged 81–90 (447%, $p < 0.001$) and in non-travel associated cases (64%, $p = 0.025$) while odds were significantly decreased in patients aged 21–30 (35%, $p = 0.046$) and in *S.* Paratyphi A infections (35%, $p = 0.01$). Isolates linked to carriage failed to cluster under specific lineages or genotypic clades, aside from a small *S.* Paratyphi B population potentially circulating locally in England. This study highlights increased prevalence of acute enteric fever infection in more deprived communities, and elevated risk of TS carriage in elderly patients and those without recent foreign travel. Our findings

**Data availability statement:** All data underlying the findings of this study are available within the manuscript and its Supporting information files. FASTQ sequences have been deposited in the National Center for Biotechnology Information (NCBI) Sequence Read Archive under BioProject accession number PRJNA248792 (www.ncbi.nlm.nih.gov/bioproject/?term=248792). S3 Table provides the individual SRA accession number, which also includes the isolate details used in the phylogenetic analysis together with the isolate level metadata permissible under the General Data Protection Regulation (GDPR).

**Funding:** AMN was supported by the Medical Research Council and the Microbes, Microbiomes and Bioinformatics Doctoral Training Partnership, grant number MR/W002604/1. The funders had no role in study design, data collection and analysis, decision to publish or preparation of the manuscript.

**Competing interests:** The authors have declared that no competing interests exist.

suggest host immune function likely plays a greater role in carriage risk than bacterial genotype. Given the lack of genetic signature for carriage, future research must focus on host factors influencing persistence, and repeat sampling post-antibiotic treatment should be implemented to identify 'modern day typhoid Marys' and reduce disease transmission.

## Author summary

Enteric fever, caused by *Salmonella enterica* serovars Typhi and Paratyphi A/B, is a major global health issue. The burden of enteric fever is worsened by bacterial carriers – individuals who harbour the bacteria for extended periods of time, often asymptomatically – and can unknowingly spread disease. Yet little is known about carriage of typhoidal *Salmonella*. This study used public health data from England and Wales over two decades to explore disease trends and identify patient groups at risk of carriage. Epidemiological analysis revealed enteric fever incidence has increased in recent years, most associated with travel to Southern Asia and disproportionately affecting disadvantaged communities. A small proportion of patients carried the bacteria for longer than three weeks, with those of older age and those who had not recently undertaken foreign travel being significantly more likely to become carriers. Outside of a small number of *S.* Paratyphi B isolates, the bacteria themselves did not show clear genomic patterns linked to carriage, suggesting the patient's immune system may play a bigger role in carriage than the genomics of the causative organism. These findings highlight key at risk groups for both acute and carriage infections and highlight the need to improve follow-up testing to identify carriers.

## Introduction

The concept of a 'healthy bacterial carrier' first captured public attention via the infamous case of typhoid Mary in the early 1900s. Though showing no symptoms herself, Mary Mallon was a cook who unknowingly spread *Salmonella enterica* subspecies *enterica* serovar Typhi (*S.* Typhi) in the food she prepared, triggering outbreaks of typhoid fever encompassing at least 51 cases and 3 deaths [1]. As the first widely publicised instance of asymptomatic carriage, Mary's story is deeply embedded in the history of infectious disease and emphasises the danger posed by 'silent carriage'.

Over a century later, disease caused by *S.* Typhi and *S.* Paratyphi A, B and C – the causative agents of enteric fever [2] – remain a major global health issue. Enteric fever causes 9.3 million cases and 107,500 deaths annually [3], primarily transmitting through contaminated food and water [2]. Southern Asia, especially the Indian subcontinent, bears the highest disease burden, but typhoidal *Salmonella* (TS) are endemic in many low- and middle-income countries in Asia, Africa and South America [4]. In higher income countries, >85% of cases are linked to international travel [5,6],

with non-travel associated infections often stemming from infected household contacts, foreign visitors or TS carriers [7] – the modern-day typhoid Marys.

Chronic TS carriers harbour the same bacterial population for years, primarily attributed to persistent gallbladder colonisation [8,9], and sporadically excrete them in faeces, leading to onwards disease transmission. TS carriers themselves are at increased risk of site-specific disease and carcinoma [10], and although prolonged antibiotic therapy has shown some efficacy for carriage resolution, such options are increasingly limited by rising antimicrobial resistance (AMR) [11]. Carriage for >12 months is estimated to occur in 2–5% of clinical enteric fever cases [12–15], yet the intermittent nature of shedding and lack of standardised diagnostics make detection – and consequently research into carriage – a significant challenge.

In England and Wales, informal global sentinel surveillance of enteric fever is undertaken by UK Health Security Agency's (UKHSA: formerly Public Health England) Gastrointestinal Bacterial Reference Unit (GBRU) via routine bacterial identification and sequencing [16]. Enhanced surveillance is undertaken by UKHSA's Travel Health and International Health Regulations Team, using questionnaires to collate in-depth demographic and behavioural details relevant to infection [17], which are published annually online [18]. This routine testing and surveillance is crucial for passive identification of TS carriage by monitoring for recurring episodes.

To enhance our understanding of TS carriage, and its contributing factors, this study analyses the evolving epidemiology of enteric fever in England and Wales over 20 years (2004–2023) to facilitate retrospective identification of TS carriers. Data presented here is crucial for bridging critical gaps in our understanding of disease transmission, bacterial evolution and the effectiveness of interventions, ultimately supporting efforts to control and prevent future cases.

## Results

### Overview of enteric fever cases

As part of their surveillance programme, the UKHSA GBRU received 9710 TS isolates within the study period (2004–2023), predominantly isolated from blood (5922, 61%) or faeces (3021, 31%). These isolates were used to calculate invasive indices (isolates from blood: isolates from faeces) for each serovar, which were 69.3 (2400:1062), 66.6 (3398:1702) and 32.5 (124:257) for *S.* Paratyphi A, *S.* Typhi and *S.* Paratyphi B, respectively.

These isolates originated from 8335 patients with 1208 (14%) contributing more than one isolate. 25 individuals experienced coinfections with more than one TS serovar simultaneously, and 13 individuals were diagnosed with two separate infections. Thus, a total of 8297 unique enteric fever infections were available for further analysis: 4696, 3275 and 326 caused by *S.* Typhi, *S.* Paratyphi A or B, respectively. Of these, 98% (8137/8297) occurred in England, and cases were distributed across the entirety of each country.

From the 4262 patients who had travelled abroad and could confirm vaccination status, only 929 (22%) had received a vaccine within its efficacious period (3 years) while 2694 (63%) had never been vaccinated, rising to 73% (1786/2453) in *S.* Typhi cases specifically. From the 5862 patients that confirmed antibiotic usage, 5662 (97%) indicated antibiotics were prescribed to them, with 3466 patients naming at least one class. Cephalosporin use was cited by 2042 (59%) patients, followed by (fluoro)quinolones and macrolides by 758 (22%) and 669 (19%) patients, respectively. (Fluoro)quinolones saw reduced usage over the study period, dropping from 41% (102/251) of named prescriptions in 2006 to 4% (18/426) in 2023.

### Overview of typhoidal *Salmonella* carriage

In total 224 carriers were identified over the study period (2.7% of cases), with 194, 21 and 9 representing convalescent, temporary or chronic carriage, respectively. Of these, 98% (219/224) of carriers were identified in England, with no substantial local geographical clustering of carriage patients. The proportion of carriers varied between serovars (Table 1) but

**Table 1. Prevalence of carriage infections among cases of enteric fever in England and Wales, 2004-2023, split by typhoidal *Salmonella* serovar.**

| Causative Agent | Carriage Type | No. of Carriage Infections | Cases Progressing to Carriage (%) |
|---|---|---|---|
| *S.* Typhi | Total | 145 | 3.1 |
| | Convalescent | 133 | 2.8 |
| | Temporary | 6 | 0.1 |
| | Chronic | 6 | 0.1 |
| *S.* Paratyphi A | Total | 62 | 1.9 |
| | Convalescent | 51 | 1.6 |
| | Temporary | 8 | 0.2 |
| | Chronic | 3 | 0.1 |
| *S.* Paratyphi B | Total | 17 | 5.2 |
| | Convalescent | 10 | 3.1 |
| | Temporary | 7 | 2.2 |
| | Chronic | 0 | 0.0 |

The cases progressing to carriage are presented as a percentage of total case numbers per serovar, which are 4696, 3275 and 326 for *S.* Typhi, *S.* Paratyphi A and B, respectively.

all exhibited fewer carriage infections in carrier states associated with longer *in vivo* persistence. Cases of *S.* Paratyphi A had 35% lower odds of being carriage infections compared to cases of *S.* Typhi, indicating a significantly reduced risk ($p = 0.01$).

Where cases were associated with travel, only 24% (28/119) and 22% (901/4143) of patients with carriage or acute infections, respectively, had received vaccination within its efficacious period prior to infection. Thus, no significant link was identified between carriage and *S.* Typhi vaccination. Furthermore, 95% (157/166) and 97% (5505/5696) of patients with carriage and acute infections, respectively, were prescribed antibiotics and no significant link was identified between carriage and antibiotic therapy. Cephalosporins were the most prescribed antibiotic for acute and carriage infections and overall only minimal differences were seen in antibiotics prescribed to acute and carriage infections (Fig 1).

### Enteric fever case burden and trends

While UKHSA publishes annual data from England, we examined overall disease burden across the 20-year study period from both England and Wales to assess long-term trends and explore patient deprivation indices. A mean of 415 enteric fever cases were recorded per annum (Fig 2) across the study period, dropping substantially during the COVID-19 pandemic, before increasing and subsequently exceeding pre-pandemic levels, with 652 cases in 2023. Case numbers were consistently highest during the late summer months to early autumn, peaking during September before dropping to the lowest levels between October and December (S1 Fig). *S.* Typhi was the dominant serovar in all but two years, with an average of ratio of 1:1.6 *S.* Paratyphi A to *S.* Typhi cases and 1:16.7 *S.* Paratyphi B to *S.* Typhi cases.

No gender bias was observed over the 20-year period examined, with highest case prevalence (2141/8258, 26%) consistently reported in adults aged 21–30 (S2 Fig). Other patient demographics also showed limited variation across the study, with >82% of *S.* Typhi (2844/3455) and *S.* Paratyphi A (1872/2251) cases reported in patients with ethnic links to Southern Asia, contrasting to the 51% of *S.* Paratyphi B patients (108/210) who were of White British ethnicity. Integration of indices of multiple deprivation (IMD) deciles indicated a higher burden of disease in lower IMD deciles (associated with more deprived small areas in England and Wales) consistent across time and TS serovar – signifying a greater number of infections in patients living in more deprived areas (Table 2). When overlaid with ethnicity data, this trend was especially

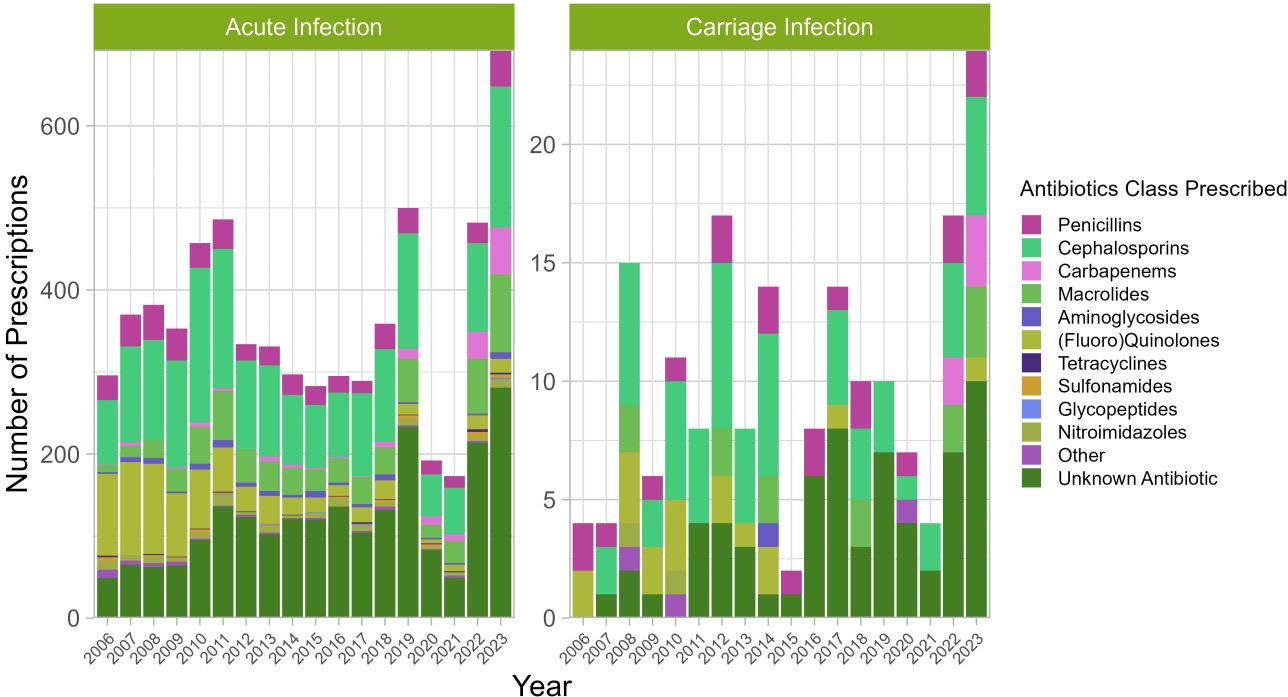

**Fig 1. Types of antibiotics used to treat acute and carriage infections of enteric fever in England and Wales, 2004-2023.** Stacked bar charts show the type of antibiotics prescribed to patients by year – total number of prescriptions exceeds patient number due to patients receiving treatment with multiple agents. Where patient could not name a specific antibiotic, a count is taken towards 'Unknown Antibiotic'.

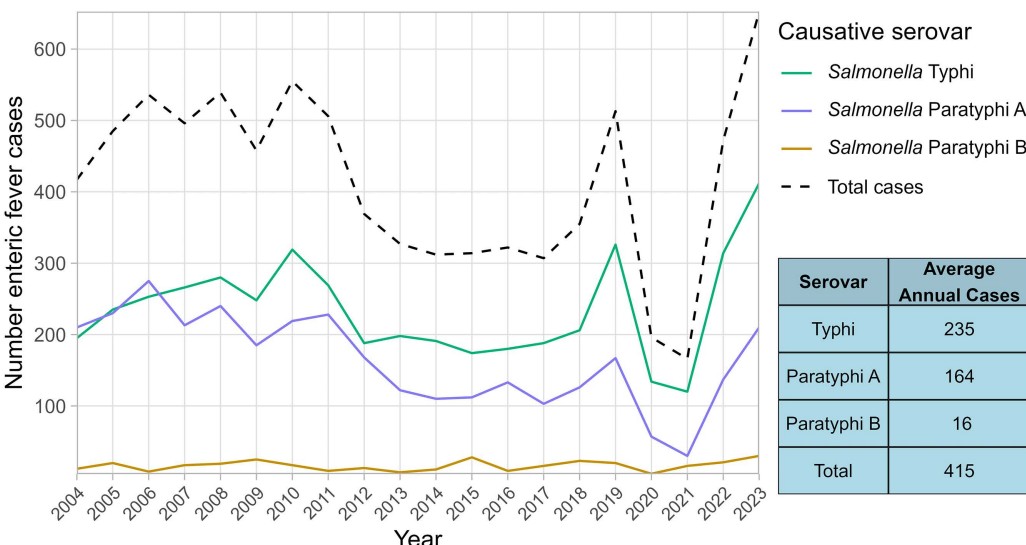

**Fig 2. Total cases of enteric fever reported in England and Wales, 2004-2023.** Total cases of enteric fever from all causative agents are shown as a black hashed line, while cases of each serovar are shown in solid lines of green, blue and orange for *S*. Typhi, *S*. Paratyphi A and *S*. Paratyphi B, respectively. A table showing the mean number of cases per annum (taken across the entire 20-year study period) for each category is also shown.

**Table 2. Distribution of enteric fever cases by indices of multiple deprivation (IMD) decile in England and Wales, 2004-2023.**

| IMD Decile | No. of Cases | Percentage of Cases (%) |
|---|---|---|
| 1 – Most deprived | 1153 | 15.9 |
| 2 | 1149 | 15.8 |
| 3 | 1140 | 15.7 |
| 4 | 907 | 12.5 |
| 5 | 699 | 9.6 |
| 6 | 657 | 9.0 |
| 7 | 462 | 6.4 |
| 8 | 358 | 4.9 |
| 9 | 390 | 5.4 |
| 10 – Least deprived | 358 | 4.9 |
| Total | 7273 | |

Table shows the raw number of cases, and percentage of overall cases, linked to each IMD decile, in infections where patient postcode was known.

pronounced in South Asian and African ethnic groups (S3 Fig). It is notable that of the 2034 Pakistani patients, 515 (25%) were classified as IMD decile 1 compared to the 44 (2%) in IMD decile 10. But conversely, incidence increased in higher IMD deciles among the 447 patients of White British ethnicity, with 10 (2%) in IMD decile 1 and 69 (15%) in IMD decile 10.

Across the study period, travel was reported in 92% (7090/7743) of cases, with yearly percentages only dropping below 90% pre-2007 and in 2021. Where cases provided a UK return/arrival date, 96% (5333/5552) reported symptom onset prior to or within 28 days of return/arrival. Travel to India and Pakistan showed the greatest association with infection, accounting for 3029 (41%) and 2454 (33%) of the 7379 potential exposure events, respectively (Fig 3). *S.* Typhi and *S.* Paratyphi A cases were linked primarily to Southern Asia, but *S.* Paratyphi B cases were more frequently linked to South America and Western Asia (S4 Fig). Visiting friends and/or relatives (VFR) was the primary travel motivation in *S.* Typhi (76%, 2629/3459) and *S.* Paratyphi A (76%, 1801/2361) cases, mainly among patients with ethnic links to the visited region. In *S.* Paratyphi B cases, travel motivations were split between VFR (48%, 103/215), primarily linked to unspecified Asian or other ethnic groups travelling to Western Asia, and holidays (45%, 97/215), mainly linked to patients of White British ethnicity travelling to South America. No significant shifts in locations visited or motivations for patient travel were identified within the study period.

### Typhoidal *Salmonella* carriage burden and trends

Over the study period, carriage infections occurred in 2.7% of cases, though annual fluctuations were observed (2007 – 1.2%, 6/496, 2017 – 6.2%, 19/307) independent of total case number (Fig 4). No new carriage infections of *S.* Paratyphi B commenced prior to 2010, or in 5 other years (2013, 2018, 2020, 2021 and 2023), and none were associated with *S.* Paratyphi A in 2007. New *S.* Typhi carriage infections commenced every year of the study.

Males accounted for 55% of cases in both acute and carriage infections (acute - 3628/7976, carriage - 124/224), and no significant link was identified between carriage and patient sex. The highest incidence of acute infection, at 26% (2099/8034), was amongst patients aged 21–30, while carriage was most prevalent, at 23% (52/224), in children aged 0–10 (Fig 5). Statistical modelling identified significant links between age and carriage: patients aged 21–30 had 35% lower odds of cases being carriage infections ($p = 0.046$) compared to the general population, while those aged 81–90 had over 447% higher odds ($p < 0.001$). Patient ethnicity did not notably vary between acute and carriage infections, and while

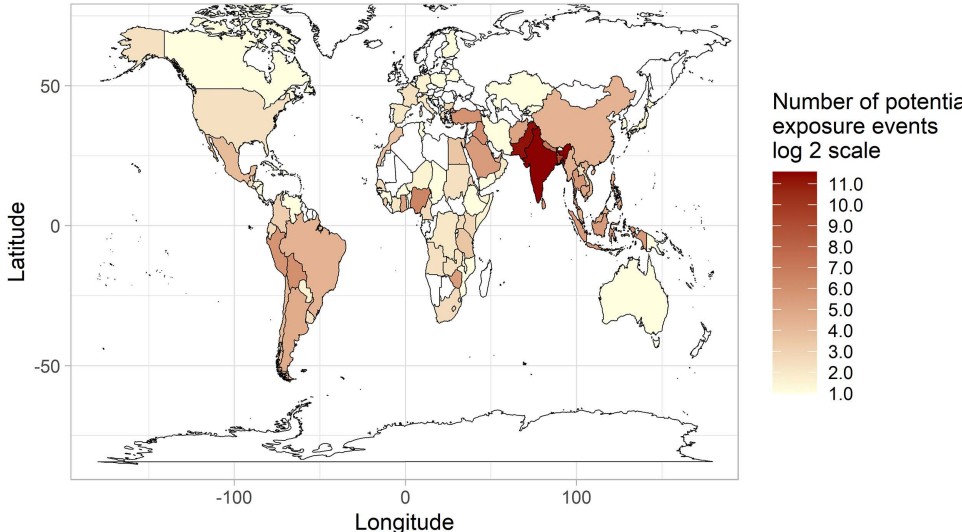

**Fig 3. Geographical origins of exposure in travel-associated enteric fever cases in England and Wales, 2004-2023.** World map shows individual countries coloured based on the number of recorded instances of travel to them in travel associated cases (potential exposure events). Colour scale is of $\log_2$, with white colouring signifying no potential exposure events, and a gradient of pale yellow to red signifying low to high numbers of potential exposure events attributed to that country. Base map: Base map boundaries were obtained from the R package maps, as 'world map', which uses the public domain 1:50m world map v2.0 'Natural Earth Project' dataset (https://www.naturalearthdata.com/), as confirmed within the documentation: (https://www.rdocumentation.org/packages/maps/versions/3.4.2).

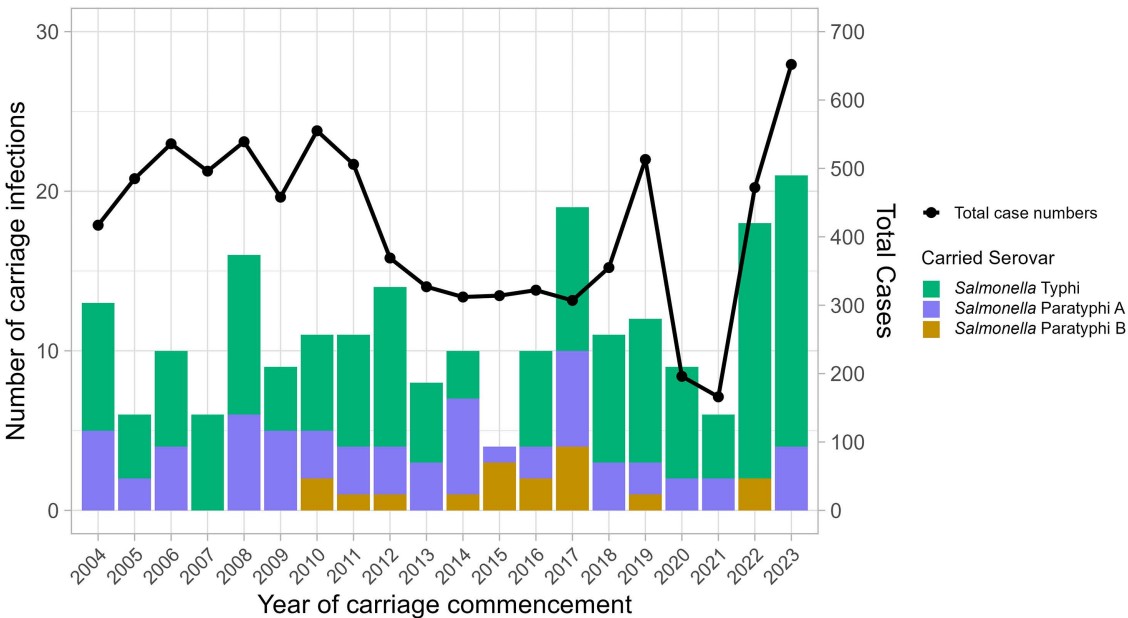

**Fig 4. Total number of carriage infections of enteric fever in England and Wales, 2004-2023.** Bars split by serovar detail number of carriage cases, for which the first isolate was obtained in the detailed year, on the leftmost scale. Overlaid line shows total number of cases, from all causative serovars each year, on the rightmost scale.

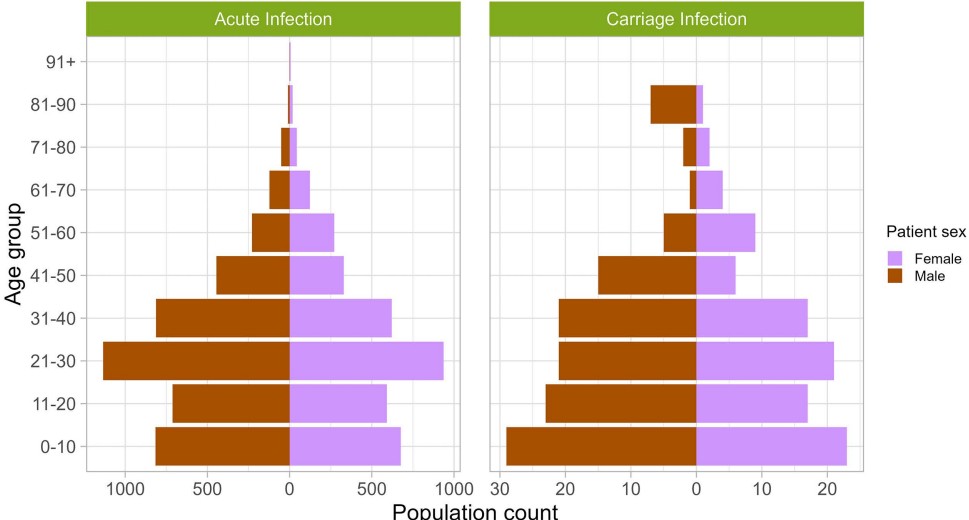

**Fig 5. Age-sex pyramid plots illustrating the distribution of patients diagnosed with acute and carriage infections of enteric fever in England and Wales, 2004-2023.** In cases where both age and sex were known, the proportion of cases observed in males (brown) and females (pink) is shown stratified by age group and split by whether the infection was classified as acute (left) or carriage (right).

lower patient IMD deciles were generally associated with more cases in acute infections, it was more variable in carriage infections, with patients of IMD decile 4 (34/205) associated with highest burden and IMD decile 7 with the least (9/205).

Travel association was less common in patients with carriage infections at 85% (184/216), compared to 92% (6906/7527) in acute infections. Among *S.* Paratyphi B carriage infections, only 53% (8/15) were travel-associated, although this is likely an artifact of the low number of cases associated with this serovar. Statistical analysis indicated individuals with no reported travel had 64% higher odds of cases being carriage infections ($p = 0.025$). Where a UK return/arrival date was specified, 95% (142/150) of carriage cases reported symptom onset prior to return or within 28 days of travel, similar to the 96% (5191/5402) in acute cases.

The geospatial distribution of potential exposure events showed minimal variation between acute and carriage infections – with both primarily associated with the Indian subcontinent. For both case types, 41% (acute – 2950/7188, carriage – 79/191) of potential exposure events linked to India, while 33% (2380/7188) and 39% (74/191) linked to Pakistan in acute and carriage infections, respectively. While all carriage infections linked to South America were of *S.* Paratyphi B, no differential exposure profiles were seen between acute and carriage infections overall (Fig 6). No differences in travel motivation were seen between case types.

### Phylogenomic analysis of typhoidal *Salmonella* from patients in England and Wales

Publicly available, high-quality genomes were available for the representative isolates in 46% (2176/4696), 36% (1168/3275) and 55% (179/326) of *S.* Typhi, *S.* Paratyphi A and *S.* Paratyphi B cases, respectively for phylogenetic analysis. Assembly availability was limited across all serovars due to routine UKHSA whole genome sequencing (WGS) only commencing in April 2014 [16], and the need for Enterobase assemblies to meet previously described quality control metrics for publication [19].

For *S.* Typhi, 12 sequence types (STs) were identified across the phylogenetic tree, with most isolates being ST1 (73%, 1578/2177) represented by large clade one and small clade three (Fig 7). ST2 accounted for 25% (547/2177) of isolates and formed clade two, which showed greater genetic diversity than clade one, by comparative branch length, and

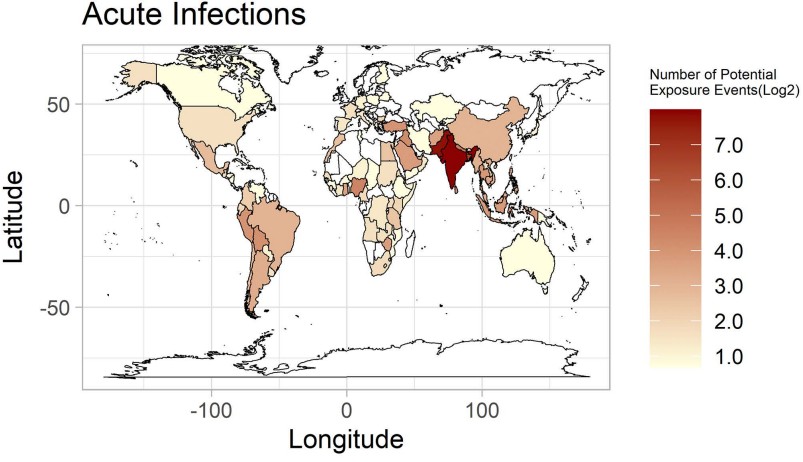

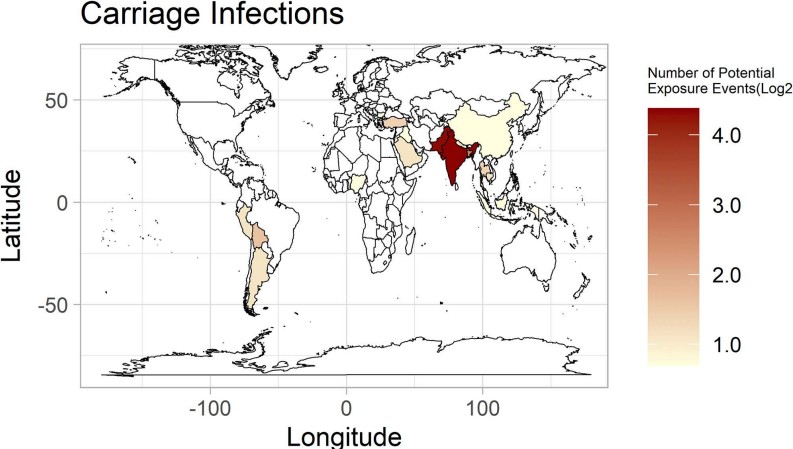

**Fig 6. Geographical origins of exposure in travel-associated acute and carriage infections of enteric fever in England and Wales, 2004-2023.**
World map shows individual countries coloured based on the number of recorded instances of travel to them in travel associated cases. Colour scale is of $\log_2$, with white colouring signifying no reported cases, and a gradient of pale yellow to red signifying low to high numbers of cases. Scale values differ between acute infections (top) and carriage infections (bottom). Each map possesses a separate scale. Base map: Base map boundaries were obtained from the R package maps, as 'world map', which uses the public domain 1:50m world map v2.0 'Natural Earth Project' dataset (https://www.naturalearth-data.com/), as confirmed within the documentation: (https://www.rdocumentation.org/packages/maps/versions/3.4.2).

encompassed other isolates of low-prevalence STs. Within the collection, 47 of the 87 Genotyphi genotypes [20] were present, with clade one aligning with genotype 4, predominantly 4.3.1.1, representing the globally disseminated H58 haplotype harbouring enhanced antibiotic resistance [21].

No STs, genotypes or locations were linked specifically to carriage infections, with carriage isolates being distributed across the entire breadth of the tree - suggesting no clear genetic signature, as determined by small nucleotide polymorphism (SNP) analysis, was present for carriage in *S.* Typhi. No carriage isolates were present in clade three, but this was likely due to the relatively small size of this cluster. Isolates linked to travel to Asia were widely present across all clades, whereas those linked to African travel formed small clusters, such as those in genotypes 4.3.1.1.EA1 and 3.1.1. Non-travel associated isolates were distributed across the tree and showed little clustering. Some temporal clustering was observed, such as a group of genotype 4.3.1 isolates between 2020–2023 (black box, Fig 7) all of which were associated

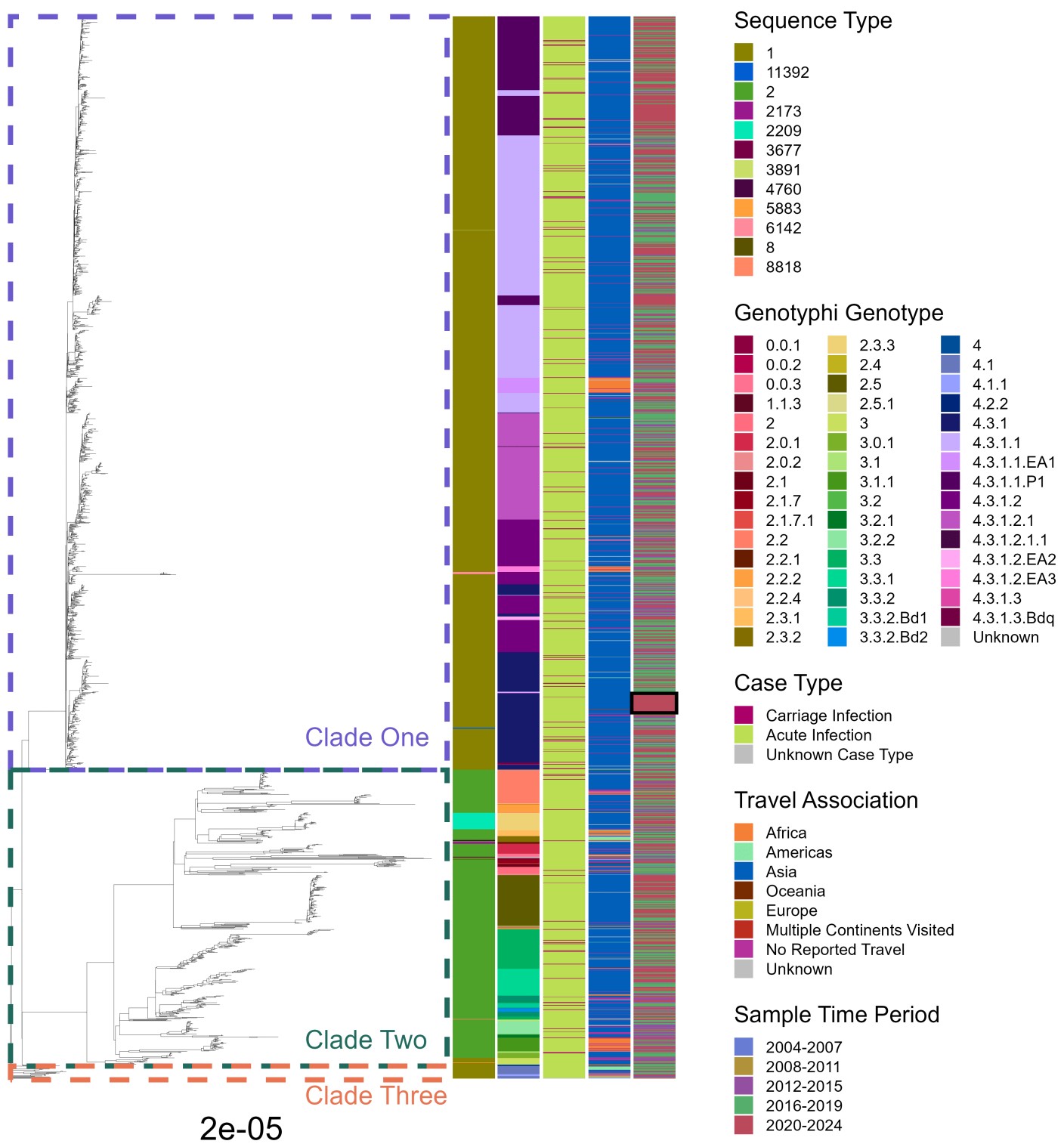

**Fig 7. Phylogenetic tree of *S.* Typhi isolates representative of enteric fever infections in England and Wales, 2004-2023.** Phylogeny tree of maximum likelihood generated using SNP-sites from a full snippy alignment with Ty2 (NCBI: ASM754v1) as a reference. Three key clades are highlighted by

coloured boxes. Heatmap bands to the right represent isolate data on sequence type (left most), Genotyphi genotype, carriage status, travel association and the period in which the sample was taken (right most). An example of temporal clustering is highlighted by a black box. Tree with bootstrapping values displayed in S5 Fig.

with India or Pakistan thus potentially corresponding to an outbreak. Genotypic diversity remained stable over the period of the study, with no evidence of lineage displacement.

For *S.* Paratyphi A, isolates were of two major STs: ST 85, in clades one and three, and ST129, forming the majority of clade two, with these STs accounting for 54% (635/1169) and 44% (510/1169) of isolates, respectively (Fig 8). Across these isolates, 19 of the 25 known Paratype genotypes [22] were present, with Paratype 2.3.3 forming a large part of clade one, while genotypes 2.4.2 and 2.4.5 were abundant in clade two. Small populations of highly genetically diverse isolates existed in both clades one and two (yellow diamonds, Fig 8) which, alongside isolates from clade three, extended a long evolutionary distance away from the population – although the reason for this is unclear.

Aside from some small clusters in Paratypes 2.3 and 2.3.4, *S.* Paratyphi A isolates from carriage infections were dispersed across the tree and seen proportionately equally in all clades. Thus, there appears to be no genetic signature for carriage in *S.* Paratyphi A that can be determined via SNP analysis. Isolates linked to Asia spanned almost all Paratypes, with a small clade of African isolates clustering in Paratype 2.3. While several isolates obtained between 2020–2024 cluster across the tree, there was no specific convergence on a single genotype of *S.* Paratyphi A over time.

All isolates of *S.* Paratyphi B were ST86, and formed three distinct clades, with clade three showing the greatest level of genetic diversity by comparative branch length (Fig 9). From a single nucleotide variant (SNV) based typing scheme [23], 14 of the 38 *S.* Paratyphi B genotypes were present with 47% (85/180) of isolates assigned genotype 10.3.6_South-America, all of which were in clade one, with the majority linked to travel to the Americas (69/85, 81%). Clade two contained isolates linked to Asia, with 10.3.2_MiddleEast2 within this clade accounting for 29% (52/180) of isolates, alongside genotype 10, 10.3.1_SouthAsia1 and 10.3.8.1_SouthAsia2. A clear geographical signal could be seen within isolates of *S.* Paratyphi B despite the serovars clonal nature, with American and Asian isolates separated into different genotypes.

Isolates from carriage infections were distributed across all of clade one and a small section of clade two. Clade three showed the greatest level of genetic diversity, with isolates grouping under genotypes 2.1 and 5. There appeared to be a slightly increased proportion of carriage infection isolates within this clade, however this may reflect the limited number of isolates in this clade rather than a true biological signal. When assessed alongside travel data, almost all cases associated with this clade were not travel associated, suggesting clade three may indeed represent a genetically distinct, domestically associated cluster within England and Wales.

## Discussion

With 415 cases of enteric fever reported on average annually in England and Wales, the requirements for treatment, contact tracing and health interventions mean this disease continues to impose a significant public health burden. Over the twenty-year period examined, patient demographics and travel associations were markedly stable, with the serovar proportion and genomic profiles of infecting organisms also exhibiting only limited change (Figs 2 and 7–9). These consistencies validate the robustness of UKHSAs surveillance system and highlight the persistence of long-established patterns in enteric fever importation. Alongside this study, surveillance data presented annually by UKHSA is pivotal for ongoing disease monitoring and essential to inform, implement and assess novel prevention strategies aimed at disrupting these deep-seated disease patterns.

UKHSA annual reports have recently documented rising enteric fever cases [18]; here 20-year longitudinal analysis confirms 2023 exhibited the highest incidence across the study period (Fig 2), surpassing pre-pandemic levels, in a trend that continued into 2024 [24]. Many endemic regions continue to report high disease burden and new outbreaks [25,26], however international travel remains below pre-pandemic levels [27] so increased case importation is unlikely without a

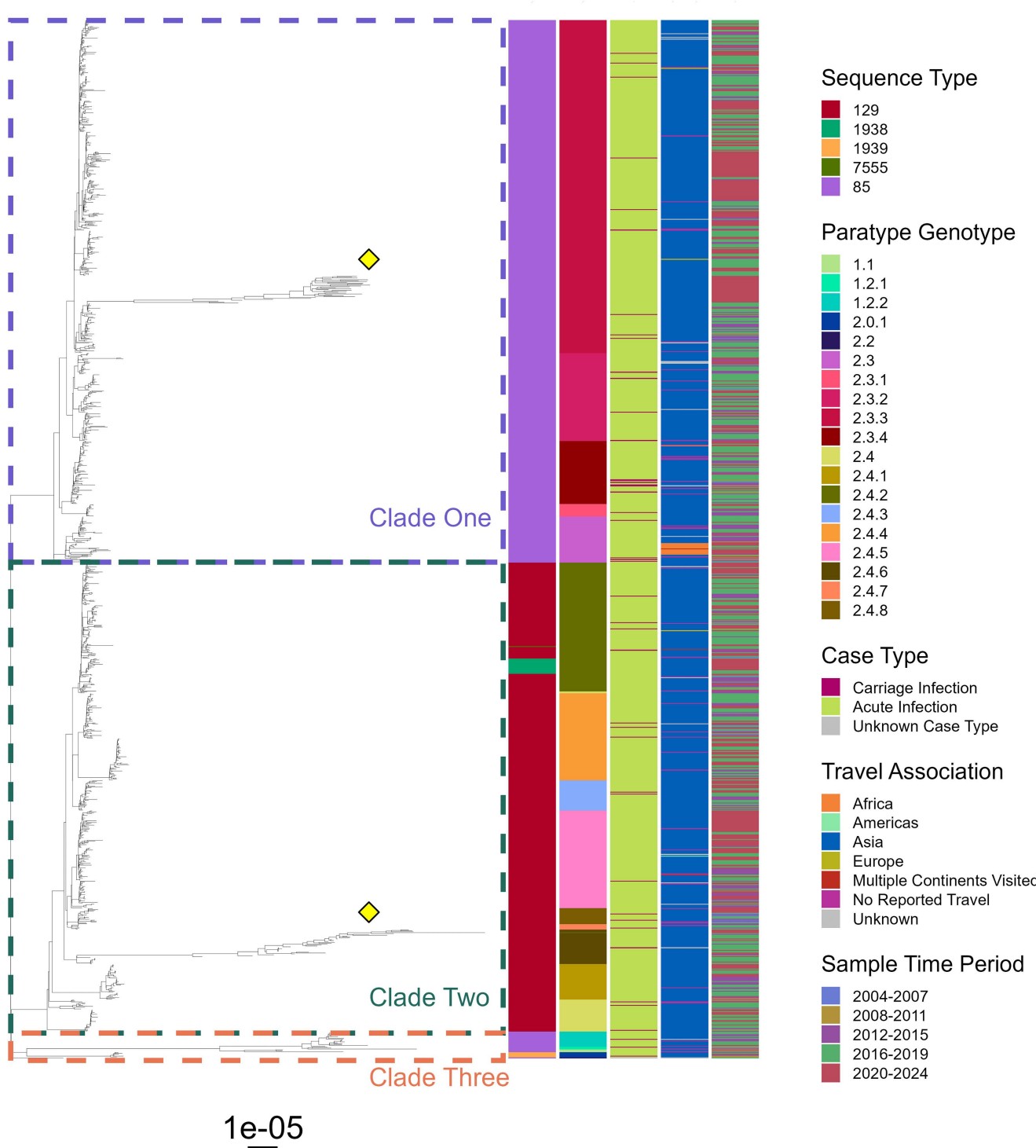

**Fig 8. Phylogenetic tree of *S*. Paratyphi A isolates representative of enteric fever infections in England and Wales, 2004-2023.** Phylogeny tree of maximum likelihood generated using SNP-sites from a full snippy alignment with ATCC 9150 (NCBI: ASM1188v1) as a reference. Three key clades are highlighted in coloured boxes, with small populations of highly diverse isolates also being highlighted (yellow diamonds). Heatmap bands to the right represent isolate data on sequence type (left most), Paratype genotype, carriage status, travel association and the period in which the sample was taken (right most). Tree with bootstrapping values displayed in S6 Fig.

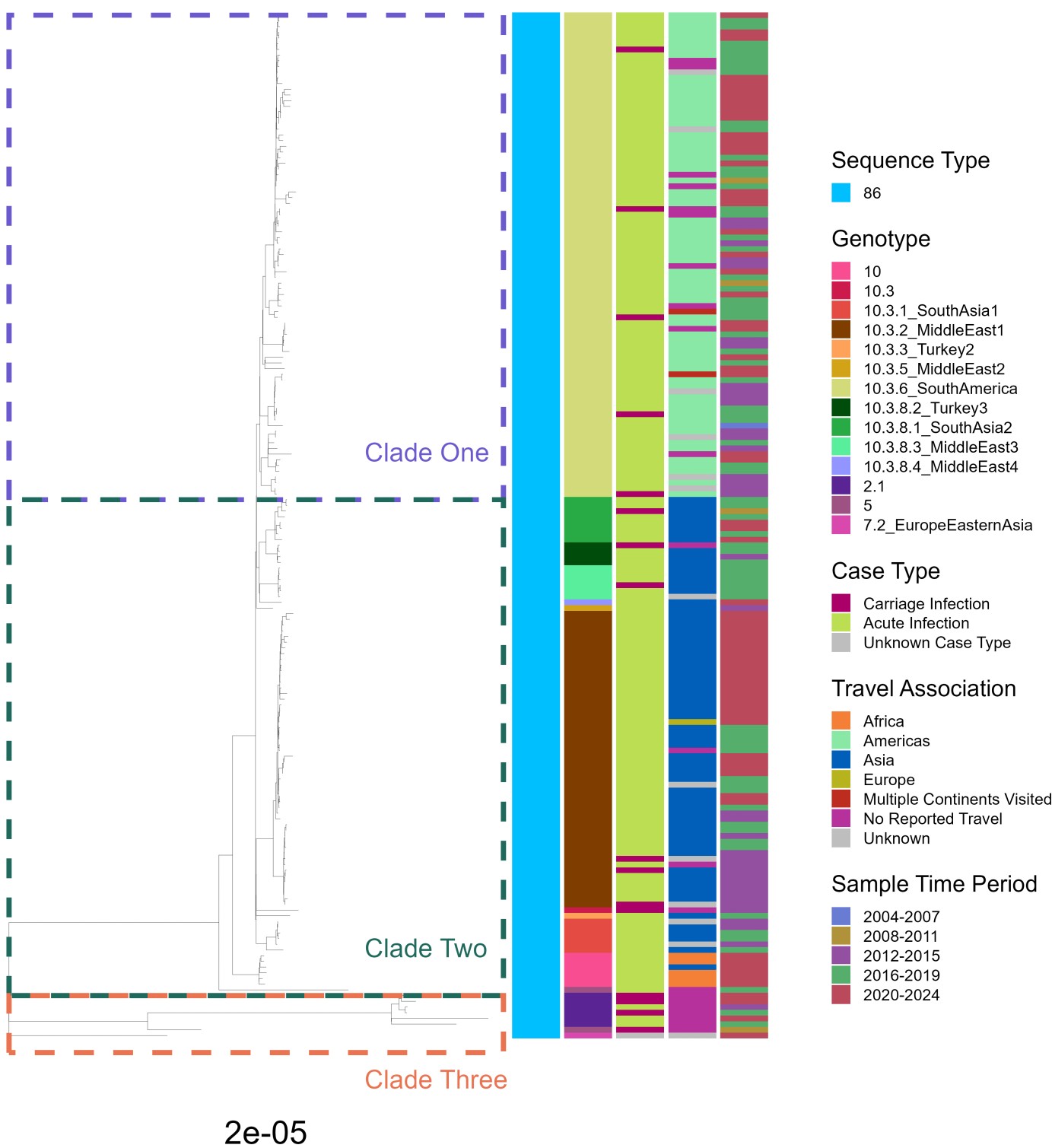

**Fig 9. Phylogenetic tree of *S*. Paratyphi B isolates representative of enteric fever infections in England and Wales, 2004-2023.** Phylogeny tree of maximum likelihood generated using SNP-sites from a full snippy alignment with B97 (NCBI: ASM3977815v1) as a reference. Two major clades are highlighted on the tree in coloured boxes. Bands to the right represent isolate data on sequence type (left most), genotype, carriage status, travel association and the period in which the sample was taken (right most). Tree with bootstrapping values displayed in S7 Fig.

rise in endemic incidence – the potential occurrence of which is difficult to ascertain given the limited nature of global surveillance. The lack of convergence on TS genotypes (Figs 7–9) also suggest emergence of a new lineage is unlikely to be a contributing factor, hence the exact underlying cause for this increased disease incidence is unknown. Increased patient vulnerability could play a contributing role to these increases, with pandemic-related social restrictions and subsequent behavioral shifts [28,29] potentially reducing bacterial exposure within the population, leading to reduced microbiome diversity. This could impair the development or maintenance of protective host-microbiome interactions and thus increase infection susceptibility [30,31]. Further research is needed to determine the true cause of increased infections and emphasises the importance of continued UKHSA surveillance.

Patients associated with lower IMD deciles were associated with increased prevalence of enteric fever infection (Table 2), especially in those of Asian and African ethnicity (S3 Fig) travelling to visit friends and relatives. Those from socially disadvantaged populations are less likely to seek preventative care, such as vaccinations [32], and may favor lower cost street food and beverages while travelling, which have been linked to increased disease incidence [33,34]. Conversely, increased infection prevalence in White British travelers were associated with higher IMD deciles (S3 Fig), often involving holiday travel. This may reflect an increased capacity for recreational international travel in those of more affluent backgrounds, potentially alongside differing priorities for travel. Thus, two key groups have been identified for future intervention campaigns emphasising safe food consumption and vaccination prior to travel. For all findings described here, it should be noted that small differences are present between the IMD decile categorisation criteria in England versus Wales, and that the IMD decile system only provides a general measure of deprivation at the postcode level and may not accurately reflect individual circumstances [35,36].

Chronic carriage of TS (persistence >12 months) is believed to occur following 2–5% of infections [12–15], but data from this study suggests this estimation more accurately aligns with overall carriage rates (persistence >3 weeks) (Table 1), with a much lower value of 0.1% of carriage infections being chronic. Differences in these estimates are likely influenced by several factors; original estimates were generated during the early 1900s, when enteric fever had much greater prevalence, from regions which were, at the time, endemic for disease thus representing populations with different risk factors to those in modern non-endemic regions such as England and Wales. The data presented here may underestimate TS carriage due to reliance on repeated isolates from patients. Carrier identification requires individuals to present to healthcare services which introduces potential surveillance bias, with more socially disadvantaged groups tending to present later in the course of illness [32], and as presentation is often driven by symptoms, asymptomatic carriers are likely to be missed. Furthermore, the intermittent nature shedding during carriage means organisms may not be detected within samples from a carrier individual. Nevertheless, repeat isolation remains the most feasible approach for identification within surveillance systems.

A study by Levine et al (1982) implicated high carriage rates in females >40 years of age [12], yet this study shows no bias towards sex or increased prevalence in middle age (Fig 5). Children aged 0–10 were instead identified as the most frequent carrier patients, alongside significantly increased odds of carriage in those aged 81–90 – groups likely possessing reduced immune capacity due to naivety and immunosenescence, respectively [37–39]. Given those aged 21–30, likely possessing resilient immune systems, show significantly decreased odds of TS carriage, we hypothesise that the immune capacity of a patient is likely a key driving factor regarding whether an infection can be cleared or persists long term. Studies on opportunistic pathogens have linked immune dysfunction to the impairment of colonisation clearance, including increased *Staphylococcus aureus* carriage in HIV positive patients [40], and significantly increased odds of *Staphylococcus pneumoniae* colonisation and extended colonisation episodes in children <5 years of age [41] – further evidencing links between carriage and the host immune system.

Research and guidance for enteric fever focuses almost exclusively on *S.* Typhi [42], with extrapolation of biological factors often being deemed sufficient for *S.* Paratyphi A treatment due to this serovar causing similar symptomatic disease [43]. Analysis in this study however adds to the growing wealth of research highlighting key differences between these

TS serovars [44–46], by evidencing that cases caused by *S*. Paratyphi A have significantly reduced odds of being carriage infections, compared to those of *S*. Typhi. Although an underlying reason for this is yet to be determined, targeted research on *S*. Paratyphi A could uncover factors limiting carriage capacity, which could potentially be utilised to decrease the burden of carriage in other TS serovars - highlighting the need for research on *S*. Paratyphi A as a unique pathogen.

This study identified significantly increased odds of carriage infections in non-travel associated cases, a finding which may again link to the host immune system. Individuals in England and Wales are unlikely to encounter TS in day-to-day life, hence levels of background immunity will be low within the population. This could mean when TS transmit locally, be it from TS carriers or visitors/household members returning from foreign regions while infected, immune naivety makes persistent infection more likely. Previous UKHSA investigations have noted that 12% of cases initially classified as non-travel associated can be linked to foreign travel when the assessment period was extended to >60 days [7]. This suggests that some individuals may have been infected abroad, and remained asymptomatic upon their return, before developing symptoms later, leading to delayed disease reporting. As case details are not always available beyond 60 days, relevant travel information may not be captured, contributing to lower reported travel in carriers. This was exemplified in a local transmission report that described household spread where contacts were infected with genetically distinct *S*. Typhi strains, originating from a single carrier harboring a diverse bacterial population, who likely became infected over 10 years earlier in Pakistan [47]. This emphasizes the complexity of contact tracing and underscores the role that carriers can play in non-travel associated cases. Further investigation is needed to determine the full repertoire of reasons underlying increased carriage in non-travel associated cases, and it is recommended that increased prevalence in non-travel cases is considered when implementing future public health controls.

Intestinal microbiota dysbiosis caused by antibiotic therapy is a significant risk factor for persistence of non-typhoidal *Salmonella* infections (NTS) [48], however this study identified no significant link between carriage and the use of antibiotics to treat TS (S2 Table). As TS cause systemic infections and are carried in the gallbladder [8], while NTS are usually localised to the gut, changes to the gut microbiota are potentially less likely to affect TS persistence. Differences in practice may also contribute to contrasting persistence risk post-treatment, since antibiotics are routinely used for TS, but not NTS due to the gastroenteritis they cause usually being self-limiting. Although antibiotic treatment was reported in this study, clinical outcomes and treatment response were outside the scope of the analysis; a follow-up study may be useful to ascertain whether any associations exist with carriage. Other factors with little variation between carriage and acute infections identified in this study included: the type of antibiotic used (Fig 1), patient vaccination status, patient IMD decile and travel destination or motivation, thus these are also unlikely to play a key role in TS carriage.

The distribution of carriage infection isolates across both *S*. Typhi and *S*. Paratyphi A phylogenetic trees, alongside no specific linkage to genotype, geographical region of infection or time (Figs 7 and 8), suggests there is no clear genetic profile predisposing isolates to a particular case type. This aligns with data showing persistently nasally carried strains of *S. aureus* were distributed among acute clinical strains in phylogenetic analyses [49]. However, while carriage isolates spanned the entire *S*. Paratyphi B phylogenetic tree, a slightly elevated number of cases appeared in the non-travel associated isolates of genotypes 2.1 and 5 (Fig 9). This group formed a distinct cluster separated from cases associated with foreign travel, and may represent a domestically associated cluster that has circulated within UK carriers from as early as 2008. The potential predisposition of these isolates to prolonged persistence could be having a substantial impact on local disease burden, however further investigation is required to identify if this observation is coincidence or represents trackable local circulation.

In consideration of all factors described above, we conclude that, outside of a small number of *S*. Paratyphi B isolates requiring further investigation, there is no discrete genetic signature or predisposing profile for TS that will trigger carriage infections. It is instead suggested that host factors, specifically those related to the immune system, may have a greater influence on bacterial persistence. Thus, future research regarding bacterial carriage should likely shift focus onto direct examination of carrier individuals and the host factors associated with the phenomena.

The existence 'modern day typhoid Marys' highlights the need for robust enteric fever surveillance in low-incidence settings, to prevent local outbreaks. Furthermore, the lack of genetic markers for TS carriage emphasises the importance of identifying carriers via repeat bacterial isolation, and contact tracing with positive cases. Currently, UKHSA guidelines suggest that only patients in high-risk groups, such as children, food-handlers and health care workers, should continue to submit samples post-antibiotic treatment to monitor for potential carriage [50]. While expanding these guidelines to encourage all patients to continue submitting samples may be impractical, we recommend incorporating additional risk groups discussed in this study, including older adults, immunosuppressed individuals and those without recent travel history, for continued sample submission and detailed infection follow up. This would not only allow for widespread identification of 'modern day typhoid Marys', facilitating interventions to reduce transmission, but would also allow for more comprehensive exploration of the characteristics associated with carriage so that the condition can be tackled more effectively in the future.

## Methods

### Ethics statement

This study used anonymised samples and data collected through routine public-health surveillance conducted under the statutory functions of the UK Health Security Agency (UKHSA). In accordance with Health Research Authority (HRA) guidance, public-health surveillance systems do not require REC review, and anonymised surveillance data may be used for research without separate ethical approval. UKHSA is legally authorised to process confidential patient information for surveillance of notifiable infectious diseases without consent under statutory powers. All data were fully anonymised prior to analysis; therefore, individual consent was not required.

### Isolation, sequencing and serotyping

Isolates were received by UKHSA's GBRU from local diagnostic laboratories in England and Wales. Invasive index was estimated for each TS serovar using the ratio of isolates recovered from blood compared to those from faeces [51]. Prior to 2014, serotyping and biochemical tests were used for serovar identification. Post 2014, WGS with Illumina technology was routinely conducted for all *Salmonella* isolates, alongside analysis as previously described [16], with serovars confirmed *in silico* with MLST [52,53] and SeqSero [54] – using ST86 assignment specifically to type *S.* Paratyphi B. SnapperDB was utilised to determine SNP addresses at 7 levels utilising hierarchical single-linkage clustering on a matrix of pairwise SNP differences [55]. Fastq sequences were deposited in the National Center for Biotechnology Information (NCBI) Sequence Read Archive under the BioProject accession number PRJNA248792 (www.ncbi.nlm.nih.gov/bioproject/?term=248792).

### Isolate inclusion

All isolates identified as positive for *S.* Typhi, *S.* Paratyphi A or *S.* Paratyphi B referred between 1 January 2004 and 31 December 2023 were included for analysis. Date of receipt by GBRU was recorded for all cases, along with sample date where available. Where sample date was missing, dates were estimated to a 95% confidence interval using the time interval between sample collection and arrival at GBRU from cases of the same sample type (i.e., blood or faeces) with complete records. *S.* Paratyphi C infections were not included in this study as they were incredibly rare and only one case in the past 15 years was associated with carriage and has been described previously [56].

### Metadata availability

Metadata associated with cases from England and Wales was collated from isolate request forms submitted to the GBRU, and enteric fever enhanced questionnaire data [17]. Data included: sex, date of birth, ethnicity (2021 UK census groupings

[57]), home postcode, date of symptom onset, whether a case was associated with foreign travel, including details of the visited continental region down to, where possible, country (M49 United Nations Geoscheme [58]) and, where applicable, the arrival/return date to the UK from abroad. Travel associated cases were determined in line with UKHSA operational guidelines [59]. Under these, cases with symptom onset prior to return from or within 28 days of foreign travel to an endemic region are considered travel associated. Cases with onset 28–60 days post travel to an endemic region may also be assessed to determine whether travel was relevant to the infection. Cases with travel reported >60 days before onset, or to non-endemic regions, may also be classified as travel associated based on specific case details, in line with public health guidelines. Metadata regarding *S.* Typhi vaccination status and antibiotic regimes taken for enteric fever treatment were available from 2006. The English and Welsh indices of multiple deprivation 2019 [35,36] determined patients' IMD decile based on their home postcode: IMD decile 1 and IMD decile 10 correspond to the most and least deprived areas, respectively. Where patients travelled to multiple locations, gave multiple reasons for travel or were prescribed multiple classes of antibiotic, a count was taken for each – henceforth known as 'potential exposure events', 'travel motivations' and 'antibiotic prescriptions', respectively. Vaccination status refers to use of the Vi or Ty21a vaccine against typhoid fever, with the efficacious period defined as 3 years [60].

## Acute and carriage infection definitions

Three key criteria were used to define case type in each patient (Fig 10): (1) the serovar(s) of isolate(s); (2) the time period of persistence - with the first positive isolate defining confirmed disease onset and last positive isolate defining resolution, and 6 months estimated as the minimum time between potential re-exposure based on ABTA reporting average UK holiday travel peaking at 1.9 times per year [61]; and (3) the genetic differences between isolates from the same patient – with SNP address differences not exceeding the 5-SNP threshold deemed genetically similar [62,63]. Where sequencing data was unavailable, it was assumed that isolates fell within the 5-SNP threshold.

Acute infections were defined as persistence ≤3 weeks (21 days). Carriage infections were further characterised into: convalescent carriage (>3 weeks- ≤ 3 months), temporary carriage (>3- ≤ 12 months) and chronic carriage (>12 months) [5,8]. Where sample date was estimated, if multiple statuses could be assigned within the 95% confidence interval the more conservative status (i.e., less time *in vivo*) was used for definition.

## Data deduplication

To facilitate statistical analysis, patients with coinfection or multiple infections were noted in the case numbers but removed from subsequent investigation, and the dataset was de-duplicated to WGS and metadata from the first isolate from each patient. For stable' metadata (patient sex, date of birth and ethnicity) consensus across all isolates from a patient was confirmed and applied to the representative isolate. For metadata with potential to vary (home postcode, health protection team, vaccination status, travel history and motives and antibiotics taken), if details were collected within six months of the representative isolate, they were assumed to be accurate and overlaid on the representative isolate metadata.

## Data analysis

Data analysis was carried out in Excel and R studio (2024.04.2 Build 764) [64] with R (v4.3.2) [65] using tidyr (v1.3.1) [66] and dplyr (v1.1.4) [67], and figures were generated using ggplot2 (v3.5.1) [68], maps (v3.4.2) [69], gridextra (v2.3) [70] and cowplot (v1.1.3) [71]. Statistical analysis was carried out in STATA (v18) [72]. Within a single model, association between carriage infections and the following patient factors were examined: sex, age group, travel association, *S.* Typhi vaccination status, use of antibiotic therapy and the causative serovar. The model also included a restricted cubic spline, based on the sample month (1–240) for each case's representative isolate, to assess non-linear trends over the

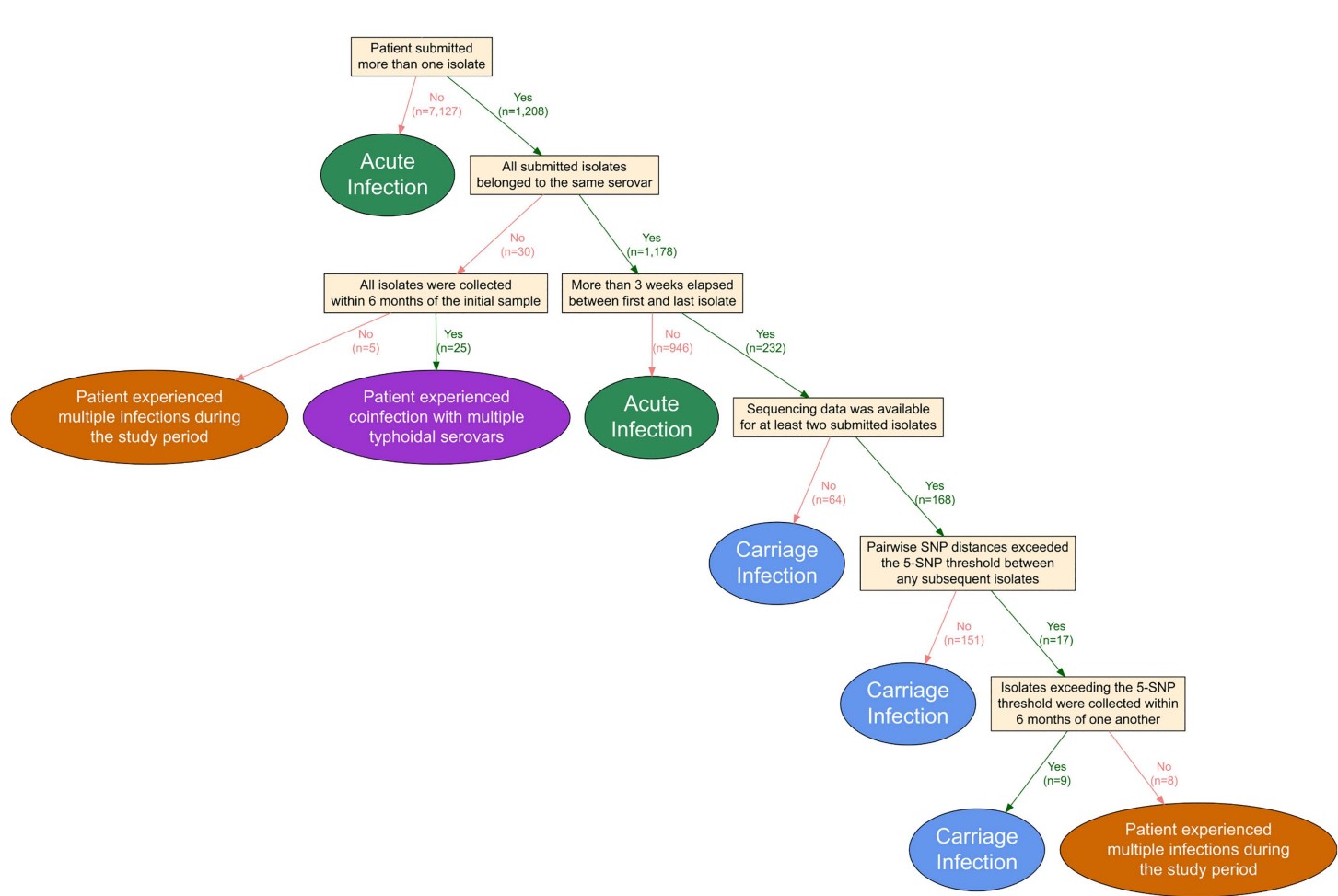

**Fig 10. Workflow for case definition.** All isolates from individual patients were used to assign a case type, based on the criteria described (yellow boxes): acute infection (persistence ≤3 weeks - green), carriage infections (persistence >3 weeks - blue), coinfection (persistence of more than one TS serovar simultaneously – purple) or multiple infections (distinct bacterial populations obtained independently – orange). The number of cases is defined at each step.

20 year-period, with knots at 12, 93 and 174 months. On categories with missing variables (S1 Table) multiple imputation using chained equations was performed 50 times, taking inference from all other factors in the model. Data was assessed for significance using a logistical regression model with unrestricted fraction missing information model test on 8297 observations, and odds ratios were collated for the above factors, using Rubin's rules, with a *p* value <0.05 considered statistically significant. Full outcomes of statistical analysis are shown in S2 Table.

## Map visualisation

Maps were generated in R using the packages 'maps' (https://cran.r-project.org/web/packages/maps/index.html) via ggplot2 so there are no copyright issues with their utilisation. Base map boundaries are obtained from the R package maps, as 'world map', which uses the public domain 1:50m world map v2.0 'Natural Earth Project' dataset (https://www.naturalearthdata.com/), as confirmed within the documentation: (https://www.rdocumentation.org/packages/maps/versions/3.4.2).

## Phylogenetics

Publicly available assemblies for representative isolates were downloaded from Enterobase (https://enterobase.warwick.ac.uk/) via SRR accession number. To determine a suitable reference for each serovar, RefSeq Masher (v0.1.2) [73–75] was used alongside visual inspection of trees containing a subset of isolates, facilitating the selection of references Ty2 (NCBI: ASM754v1) for *S.* Typhi, ATCC9150 (NCBI: ASM1188v1) for *S.* Paratyphi A and B97 (NCBI: ASM3977815v1) for *S.* Paratyphi B. Isolate SNPs against the serovar reference were defined using snippy (v4.6.0) [76], with snippy-core used to generate a full whole genome alignment, and SNP-sites (v2.5.1) [77] to remove non-informative SNPs. Maximum likelihood phylogenetic trees were generated using IQTree2 (v2.3.4) [78], with identical sequences maintained and models selected for best fit by modelfinder. Branch support values were estimated with 1000 replicate standard ultrafast bootstrapping.

Tree visualisation was carried out in R, using ggplot2 (v3.5.1) [68], ggtree (v3.10.1) [79], ggnewscale (v0.5.0) [80], tidyverse (v2.0.0) [81] and ape (v5.8) [82], with metadata overlaid using gheatmap(). Isolate ST was determined with MLST (v2.23.0) [52] in conjunction with UKHSA data. Serovar specific typing was undertaken using Paratype (v1.1) [22] for *S.* Paratyphi A, and Mykrobe (v0.13.0) [83] with panels 20240407 and 20230627 corresponding to the Genotyphi scheme for *S.* Typhi [20,84] and a single nucleotide variant SNV-based genotyping scheme for *S.* Paratyphi B [23,85], respectively.

## Supporting information

**S1 Table. Overview of variables and missing data included in statistical analysis.**
(PDF)

**S2 Table. Raw results from logistical regression model for statistical analysis of factors contributing to carriage of typhoidal *Salmonella*.**
(PDF)

**S3 Table. Metadata for isolates included in phylogenetic analysis.**
(XLSX)

**S1 Fig. Total cases of enteric fever reported by quarter in England and Wales, 2004–2023.** Total cases of enteric fever from all causative agents are shown as a black hashed line, while cases of each serovar are shown in solid lines of green, blue and orange for S. Typhi, S. Paratyphi A and S. Paratyphi B, respectively. Points for values at each quarter are given at the following dates each year: Q1 = 01/01, Q2 = 01/04, Q3 = 01/07 and Q4 = 01/10.
(TIF)

**S2 Fig. Age-sex pyramid plot illustrating the distribution of patients diagnosed with enteric fever in England and Wales, 2004–2023.** In cases where both age and sex were known, the proportion of cases observed in males (brown) and females (pink) is shown stratified by age group.
(TIF)

**S3 Fig. Relationships between patient ethnicity and indices of multiple deprivation (IMD) decile in enteric fever cases in England and Wales, 2004–2023.** Bar charts split by patient ethnicity, according to the 2021 UK census shows IMD decile distribution across patients as a percentage, with IMD decile 1 being the most deprived and IMD decile 10 being the least. The number of patients corresponding to each ethnicity is in the title of each panel.
(TIF)

**S4 Fig. Geographical origins of exposure in travel-associated enteric fever cases by causative agent in England and Wales, 2004–2023.** World map shows individual countries coloured based on the number of recorded instances of travel to them in travel associated cases, when the causative agent was S. Typhi (Top), S. Paratyphi A (Middle) or S.

Paratyphi B (Bottom). Colour scale is of log$_2$, with white colouring signifying no reported cases, and a gradient of pale yellow to red signifying low to high numbers of cases. Each serovar possesses a separate scale. Base map: Base map boundaries were obtained from the R package maps, as 'world map', which uses the public domain 1:50m world map v2.0 'Natural Earth Project' dataset (https://www.naturalearthdata.com/), as confirmed within the documentation: (https://www.rdocumentation.org/packages/maps/versions/3.4.2).
(TIF)

**S5 Fig. Phylogenetic tree of S. Typhi isolates representative of enteric fever infections in England and Wales, 2004–2023, with branch support values.** Phylogeny tree of maximum likelihood generated using SNP-sites from a full snippy alignment with Ty2 (NCBI: ASM754v1) as a reference displaying branch support values as calculated from 1000 replicate ultrafast bootstrapping.
(TIF)

**S6 Fig. Phylogenetic tree of S. Paratyphi A isolates representative of enteric fever infections in England and Wales, 2004–2023, with branch support values.** Phylogeny tree of maximum likelihood generated using SNP-sites from a full snippy alignment with ATCC 9150 (NCBI: ASM1188v1) as a reference displaying branch support values as calculated from 1000 replicate ultrafast bootstrapping.
(TIF)

**S7 Fig. Phylogenetic tree of S. Paratyphi B isolates representative of enteric fever infections in England and Wales, 2004–2023, with branch support values.** Phylogeny tree of maximum likelihood generated using SNP-sites from a full snippy alignment with B97 (NCBI: ASM3977815v1) as a reference displaying branch support values as calculated from 1000 replicate ultrafast bootstrapping.
(TIF)

## Acknowledgments

The authors would like to acknowledge the UKHSA Health Protection Teams and Environmental Health Officers for routinely completing enhanced questionnaires, and the UKHSA clinical laboratories for isolating, testing and sequencing enteric fever samples. Acknowledgements to the clinical team, in particular Gauri Godbole and Vanessa Wong within the Gastrointestinal Infection Food Safety and One Health for providing clinical advice with enteric fever complex cases. Thanks also to John Wain, Quadram Institute, for helping with the writing process. EVW and GL would like to gratefully acknowledge the support of the Biotechnology and Biological Sciences Research Council (BBSRC), and general funding provided by the BBSRC Institute Strategic Programme Microbes in Food Safety BB/X011011/1 and their constituent projects BBS/E/QU/230002B. AMN would like to gratefully acknowledge access to Quadram Institute Cloud computational resources made possible through the BBSRC Core Capability Grant BB/CCG2260/1. Patient metadata and sequence data utilised in this project originated from the UKHSA enteric fever sentinel surveillance scheme funded by UKHSA with additional support from the Health Protection Research Unit in Gastrointestinal Infections (HPRU GI) and Public Health Genomics (HPRU PHG).

The findings and conclusions in this report are those of the authors and do not necessarily represent the official position of UKHSA or Quadram Institute Bioscience.

## Author contributions

**Conceptualization:** Marie A. Chattaway.

**Data curation:** Alice M. Nisbet, David Powell.

**Formal analysis:** Alice M. Nisbet, Neville Q. Verlander.

**Funding acquisition:** Gemma C. Langridge, Marie A. Chattaway.

**Investigation:** Alice M. Nisbet, Iman Mohamed, David Powell.

**Methodology:** Alice M. Nisbet, Neville Q. Verlander, Marie A. Chattaway.

**Supervision:** Gemma C. Langridge, Marie A. Chattaway.

**Visualization:** Alice M. Nisbet.

**Writing – original draft:** Alice M. Nisbet, Emma V. Waters, Gemma C. Langridge, Marie A. Chattaway.

**Writing – review & editing:** Alice M. Nisbet, Iman Mohamed, Neville Q. Verlander, David Powell, Emma V. Waters, Andrew Nelson, Hilary Kirkbride, Gemma C. Langridge, Marie A. Chattaway.

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
