## [Decision Letter · Decision Letter 0]

22 Feb 2026

PNTD-D-25-01911

The typhoid Mary legacy: genomic epidemiology uncovers contemporary carriage dynamics across two decades of enteric fever surveillance in England and Wales

Dear Dr., Chattaway,

Thank you for submitting your manuscript to PLOS Neglected Tropical Diseases. After careful consideration, we feel that it has merit but does not fully meet PLOS Neglected Tropical Diseases's publication criteria as it currently stands. Therefore, we invite you to submit a revised version of the manuscript that addresses the points raised during the review process.

* A letter that responds to each point raised by the editor and reviewer(s). You should upload this letter as a separate file labeled 'Response to Reviewers'. This file does not need to include responses to any formatting updates and technical items listed in the 'Journal Requirements' section below.'. This file does not need to include responses to any formatting updates and technical items listed in the 'Journal Requirements' section below.

* A marked-up copy of your manuscript that highlights changes made to the original version. You should upload this as a separate file labeled 'Revised Manuscript with Track Changes'.'.

* An unmarked version of your revised paper without tracked changes. You should upload this as a separate file labeled 'Manuscript'.'.

We look forward to receiving your revised manuscript.

Kind regards,

Sanjai Kumar

Guest Editor

Stuart Blacksell

Section Editor

Shaden Kamhawi

co-Editor-in-Chief

Paul Brindley

co-Editor-in-Chief

**Additional Editor Comments:**

Please address the following comments and submit a revised manuscript for review.

Reviewer 1:

This manuscript titled “The typhoid Mary legacy: genomic epidemiology uncovers contemporary carriage dynamics across two decades of enteric fever surveillance in England and Wales” by Nisbet et al., represents significant scientific merit with important public health implications. Exceptional 20-year surveillance dataset (2004-2023) with 8,297 cases. Authors presented novel findings about carriage dynamics make substantial contributions to the field.

There are some comments to improve the manuscript:

1. Provide better justification for the 3-week threshold and discuss limitations

2. Authors needs to address methodological limitations around carriage definition and potential surveillance biases

3. Include more discussion of immunological factors that might explain age-related carriage patterns

4. Fig 1 showed data for 2006-2023. Is there any reason 2004-2005 data were excluded?

5. Please check typographical errors, like in Line 505: "Welsh"(less...)

Reviewer 3:

This manuscript examines enteric fever surveillance data from England and Wales spanning 2004-2023, with a focus on understanding bacterial carriage dynamics of typhoidal Salmonella (S. Typhi and S. Paratyphi A/B). The study analyzed 8,297 cases and identified that approximately 2.7% of infections progressed to carriage (persistence >3 weeks), with only 0.1% becoming chronic carriers (>12 months). Key findings revealed that carriage risk was significantly elevated in elderly patients (ages 81-90) and those without recent foreign travel, while patients aged 21-30 and those infected with S. Paratyphi A had reduced carriage odds. Importantly, phylogenomic analysis found no distinct genetic signatures associated with carriage isolates, suggesting that host immune factors rather than bacterial genotype play a more critical role in determining whether infections persist long-term.

The manuscript is organized and written well, with no need for extensive language edits.

The study acknowledges substantial missing data across multiple variables (referenced in S1 Table), which required multiple imputation for statistical analysis. While I believe the authors used appropriate statistical methods to address this, missing data can still introduce bias and reduce the analytical power of the study overall.(less...)

Figure 3:

The 28-day travel window prior to disease onset may not capture all travel-associated cases, as the authors note that extending this to 60 days previously identified an additional 12% of cases as travel-related. This could affect the accuracy of travel versus non-travel associations with carriage.

What was the rationale for limiting the travel window to 28 days?

Figure 5:

Despite concluding that host immune factors likely play a greater role than bacterial genotype in carriage, the study lacks direct examination of patient immune status, comorbidities, or other host characteristics beyond age. The hypothesis about immune function is inferred rather than directly tested.

Are any data available that can be included (such as comorbidities, general immune status) to begin to infer additional host factors, other than age?(less...)

Figure 7:

Only 46% of S. Typhi cases, 36% of S. Paratyphi A cases, and 55% of S. Paratyphi B cases had publicly available high-quality genomes for phylogenetic analysis. This incomplete coverage could potentially miss important genetic patterns that may be associated with trends in carriage among the cases studied.

Are further studies planned to analyze these data as more genomes become publicly available? I believe this future re-analysis, combined with any additional subject data (underlying health conditions, etc. as mentioned above) will provide interesting additional data regarding carriage trends.(less...)

**Journal Requirements:**

3) Some material included in your submission may be copyrighted. According to PLOSu2019s copyright policy, authors who use figures or other material (e.g., graphics, clipart, maps) from another author or copyright holder must demonstrate or obtain permission to publish this material under the Creative Commons Attribution 4.0 International (CC BY 4.0) License used by PLOS journals. Please closely review the details of PLOSu2019s copyright requirements here: PLOS Licenses and Copyright. If you need to request permissions from a copyright holder, you may use PLOS's Copyright Content Permission form.

Potential Copyright Issues:

i) Figures S4, 3, and 6. Please (a) provide a direct link to the base layer of the map (i.e., the country or region border shape) and ensure this is also included in the figure legend; and (b) provide a link to the terms of use / license information for the base layer image or shapefile. We cannot publish proprietary or copyrighted maps (e.g. Google Maps, Mapquest) and the terms of use for your map base layer must be compatible with our CC BY 4.0 license.

4) Please provide a detailed Financial Disclosure statement. This is published with the article. It must therefore be completed in full sentences and contain the exact wording you wish to be published.

1) Please clarify all sources of financial support for your study. List the grants, grant numbers, and organizations that funded your study, including funding received from your institution. Please note that suppliers of material support, including research materials, should be recognized in the Acknowledgements section rather than in the Financial Disclosure

2) State the initials, alongside each funding source, of each author to receive each grant. For example: "This work was supported by the National Institutes of Health (####### to AM; ###### to CJ) and the National Science Foundation (###### to AM)."

3) State what role the funders took in the study. If the funders had no role in your study, please state: "The funders had no role in study design, data collection and analysis, decision to publish, or preparation of the manuscript."

4) If any authors received a salary from any of your funders, please state which authors and which funders..

5) Your current Financial Disclosure states, "The author(s) received no specific funding for this work.".

However, your funding information on the submission form indicates receiving funds from Biotechnology and Biological Sciences Research Council and National Institute for Health Research Health Protection Research Unit.

Please indicate by return email the full and correct funding information for your study and confirm the order in which funding contributions should appear. Please be sure to indicate whether the funders played any role in the study design, data collection and analysis, decision to publish, or preparation of the manuscript.

6) Please send a completed 'Competing Interests' statement, including any COIs declared by your co-authors. If you have no competing interests to declare, please state "The authors have declared that no competing interests exist". Otherwise please declare all competing interests beginning with the statement "I have read the journal's policy and the authors of this manuscript have the following competing interests"

**Reviewers' comments:**

Reviewer's Responses to Questions

**Key Review Criteria Required for Acceptance?**

**Methods**

-Are the objectives of the study clearly articulated with a clear testable hypothesis stated?

-Is the study design appropriate to address the stated objectives?

-Is the population clearly described and appropriate for the hypothesis being tested?

-Is the sample size sufficient to ensure adequate power to address the hypothesis being tested?

-Were correct statistical analysis used to support conclusions?

-Are there concerns about ethical or regulatory requirements being met?

Reviewer #1: -Yes

-Yes

-Yes

-Yes

-Yes

-No

Reviewer #2: (No Response)

Reviewer #3: Figure 7:

Only 46% of S. Typhi cases, 36% of S. Paratyphi A cases, and 55% of S. Paratyphi B cases had publicly available high-quality genomes for phylogenetic analysis. This incomplete coverage could potentially miss important genetic patterns that may be associated with trends in carriage among the cases studied.

Are further studies planned to analyze these data as more genomes become publicly available? I believe this future re-analysis, combined with any additional subject data (underlying health conditions, etc. as mentioned above) will provide interesting additional data regarding carriage trends.

**Results**

-Does the analysis presented match the analysis plan?

-Are the results clearly and completely presented?

-Are the figures (Tables, Images) of sufficient quality for clarity?

Reviewer #1: -Yes

-Yes

-Yes

Reviewer #2: (No Response)

Reviewer #3: Figure 3:

The 28-day travel window prior to disease onset may not capture all travel-associated cases, as the authors note that extending this to 60 days previously identified an additional 12% of cases as travel-related. This could affect the accuracy of travel versus non-travel associations with carriage.

What was the rationale for limiting the travel window to 28 days?

Figure 5:

Despite concluding that host immune factors likely play a greater role than bacterial genotype in carriage, the study lacks direct examination of patient immune status, comorbidities, or other host characteristics beyond age. The hypothesis about immune function is inferred rather than directly tested.

Are any data available that can be included (such as comorbidities, general immune status) to begin to infer additional host factors, other than age?

**Conclusions**

-Are the conclusions supported by the data presented?

-Are the limitations of analysis clearly described?

-Do the authors discuss how these data can be helpful to advance our understanding of the topic under study?

-Is public health relevance addressed?

Reviewer #1: -Yes

-No

-Yes

-Yes

Reviewer #2: (No Response)

Reviewer #3: While there are clear limitations to this study, as mentioned above, this analysis spanning 20 years of data in England and Wales captures a robust set of data examining long term trends, seasonal changes and the overall temporal stability of this disease (with post-pandemic incidence increase noted). Overall, this study highlights a few key public health findings, relating to socioeconomic impacts on the spread of this disease, as well as the higher disease burden in more deprived communities – informing where more intervention through vaccination may be needed. Importantly, the authors openly discuss the challenges and limitations of the data and do not overextend their conclusions.

**Editorial and Data Presentation Modifications?**

Reviewer #1: (No Response)

Reviewer #2: (No Response)

Reviewer #3: Accept

**Summary and General Comments**

Reviewer #1: This manuscript titled “The typhoid Mary legacy: genomic epidemiology uncovers contemporary carriage dynamics across two decades of enteric fever surveillance in England and Wales” by Nisbet et al., represents significant scientific merit with important public health implications. Exceptional 20-year surveillance dataset (2004-2023) with 8,297 cases. Authors presented novel findings about carriage dynamics make substantial contributions to the field.

There are some comments to improve the manuscript:

1. Provide better justification for the 3-week threshold and discuss limitations

2. Authors needs to address methodological limitations around carriage definition and potential surveillance biases

3. Include more discussion of immunological factors that might explain age-related carriage patterns

4. Fig 1 showed data for 2006-2023. Is there any reason 2004-2005 data were excluded?

5. Please check typographical errors, like in Line 505: "Welsh"

Reviewer #2: (No Response)

Reviewer #3: This manuscript examines enteric fever surveillance data from England and Wales spanning 2004-2023, with a focus on understanding bacterial carriage dynamics of typhoidal Salmonella (S. Typhi and S. Paratyphi A/B). The study analyzed 8,297 cases and identified that approximately 2.7% of infections progressed to carriage (persistence >3 weeks), with only 0.1% becoming chronic carriers (>12 months). Key findings revealed that carriage risk was significantly elevated in elderly patients (ages 81-90) and those without recent foreign travel, while patients aged 21-30 and those infected with S. Paratyphi A had reduced carriage odds. Importantly, phylogenomic analysis found no distinct genetic signatures associated with carriage isolates, suggesting that host immune factors rather than bacterial genotype play a more critical role in determining whether infections persist long-term.

The manuscript is organized and written well, with no need for extensive language edits.

The study acknowledges substantial missing data across multiple variables (referenced in S1 Table), which required multiple imputation for statistical analysis. While I believe the authors used appropriate statistical methods to address this, missing data can still introduce bias and reduce the analytical power of the study overall.

PLOS authors have the option to publish the peer review history of their article (what does this mean?). If published, this will include your full peer review and any attached files.). If published, this will include your full peer review and any attached files.

**Do you want your identity to be public for this peer review?** For information about this choice, including consent withdrawal, please see our  For information about this choice, including consent withdrawal, please see our Privacy Policy..

Reviewer #1: No

Reviewer #2: **Yes:** ANKIT PURIANKIT PURI

Reviewer #3: No

**Figure resubmission:**
---

## [Editor Report · Decision Letter 1]

20 Mar 2026

Dear Dr. Chattaway,

We are pleased to inform you that your manuscript 'The typhoid Mary legacy: genomic epidemiology uncovers contemporary carriage dynamics across two decades of enteric fever surveillance in England and Wales' has been provisionally accepted for publication in PLOS Neglected Tropical Diseases.

Best regards,

Sanjai Kumar

Guest Editor

Stuart Blacksell

Section Editor

Shaden Kamhawi

co-Editor-in-Chief

Paul Brindley

co-Editor-in-Chief

None.

---

## [Editor Report · Acceptance letter]

Dear Dr Chattaway,

We are delighted to inform you that your manuscript, "The typhoid Mary legacy: genomic epidemiology uncovers contemporary carriage dynamics across two decades of enteric fever surveillance in England and Wales," has been formally accepted for publication in PLOS Neglected Tropical Diseases.

Best regards,

Shaden Kamhawi

co-Editor-in-Chief

Paul Brindley

co-Editor-in-Chief
